# An Application of the Theory of Planned Behaviour to Predict Intention to Consume Plant-Based Yogurt Alternatives

**DOI:** 10.3390/foods10010148

**Published:** 2021-01-12

**Authors:** Sujita Pandey, Christian Ritz, Federico Jose Armando Perez-Cueto

**Affiliations:** 1Department of Food Science, University of Copenhagen, Rolighedsvej 26, 1958 Frederiksberg C, Denmark; apce@food.ku.dk; 2Department of Nutrition, Exercise and Sports, University of Copenhagen, Rolighedsvej 25, 1958 Frederiksberg C, Denmark; ritz@nexs.ku.dk

**Keywords:** plant-based yogurt alternative, theory of planned behaviour, knowledge, sensory attributes, Denmark

## Abstract

This study attempts to predict plant-based yogurt consumers’ intention to consume plant-based yogurt alternatives in Denmark employing Ajzen’s theory of planned behaviour as a theoretical framework. An online survey was conducted among 265 consumers recruited through convenience (snowball) sampling. The results from structural equation modelling analysis show that attitude (β = 0.216, *p* < 0.001), perceived behavioural control (self-efficacy) (β = 0.229, *p* < 0.001) and perceived sensory attributes (β = 0.324, *p* < 0.001) positively and significantly influenced consumers’ intention to consume plant-based yogurt alternatives. However, subjective norms (β = 0.106, *p* = 0.087) and perceived barriers (β = −0.051, *p* = 0.414) did not influence consumers’ intention to consume plant-based yogurt alternatives. Further, objective knowledge showed no significant influence on attitudes (β = 0.077, *p* = 0.242) and intention (β = −0.029, *p* = 0.603) towards plant-based yogurt alternatives. The intention (β = 0.677, *p* < 0.001) to consume plant-based yogurt alternatives showed a strong positive correlation with the behaviour to consume plant-based yogurt alternatives. The results from logistic regression indicated that socio-demographic characteristics, gender, education, income and region of residence were significantly associated with consumption of plant-based yogurt alternatives. Thus, focusing on consumers’ attitudes and self-efficacy and improving the organoleptic characteristics might promote the consumption of plant-based yogurt alternatives in Denmark and similar countries.

## 1. Introduction

A sustainable diet is defined as a diet that has a low environmental impact and contributes to food and nutrition security, thus protecting biodiversity and the ecosystem [1]. Knowledge about and demand for a sustainable diet has resulted in the demand for alternative foods [2]. As a result, there is a growing shift towards a plant-based diet [3]. Plant-based alternatives or animal-based substitutes are considered environmentally sustainable as these foods can have a reduced negative impact on the environment [4,5]. A study in Denmark [6] showed that greenhouse gas emission was 48% higher for the average diet compared to a plant-based diet. Further, due to the growing environmental impact of livestock farming, alternative protein sources such as plant-derived protein are experiencing a growing demand globally [7]. The shift towards a plant-based diet may significantly improves food sustainability and environmental impact [8,9] and also has beneficial health implications [10,11,12]. For instance, a recent meta-analysis by Fan et al. [10] found that increasing plant protein and yogurt and reducing total protein, especially animal protein, can lead to a significant decrease in risk for diabetes type 2.

In recent years, European consumers have started considering the consumption of plant-based diets and fermented plant-based foods such as tofu or other newly developed probiotic products [13], and this is therefore no longer a niche market. Apart from environmental concerns, European consumers’ concerns about food allergies, especially nut, gluten, milk protein and lactose, are increasing and avoidance of such is regarded as a healthy lifestyle. Globally, about two-thirds of the population are lactose intolerant [14,15] and according to Skripak and colleagues [16] milk protein allergy is a common problem among infants and young children. Therefore, in Denmark, around 2.5% of the population follow a vegetarian lifestyle; however, statistics on the exact number of vegans in Denmark is lacking, though the Vegetarian Society of Denmark estimates that around 35,000 Danish consumers’ are following a vegan lifestyle [17].

A previous study has shown that plant-based dairy alternatives are appealing and tasty when they are fermented [18]. Further, the same study found that the fermentation of plant-based dairy alternatives can potentially improve sensory attributes, nutritional quality, and texture. Plant-based yogurt alternatives are a good example of fermented milk products that are based on different sources such as cereals, legumes, nuts, seeds or pseudo-cereal. A study [19] revealed that yogurt developed through the combination of coconut and soy milk results in a good source of amino and fatty acids, helps boost the immune system and can satisfy consumers’ organoleptic needs.

Regarding sensory perception and consumer acceptability of plant-based yogurt alternatives, a study found that measurable, objective and sensory attributes such as sweet, moist, soft and smooth influenced consumer product acceptance. Further, extrinsic attribute information disclosure such as ‘high in protein’ or ‘source of protein’ could help change consumer eating habits [20]. However, no research has been conducted to study the factors influencing consumer intention to consume plant-based yogurt alternatives using the theory of planned behaviour (TPB). Existing data have mainly investigated milk substitutes [21]. Thus, the underlying factors remain unclear, and more consumer research is needed to determine the key drivers of and barriers to consuming plant-based yogurt alternatives. The main aim of this study is to predict factors influencing intention to consume plant-based yogurt alternative products among plant-based yogurt consumers in Denmark. Further, the study also investigates the association between the consumption behaviour of plant-based dairy alternative products and socio-demographic characteristics and lifestyle determinants. The knowledge from this study contributes to and extends further understanding of consumers’ consumption behaviour, identifying the rationales for consuming plant-based yogurt alternatives. Hence, it might precipitate the adoption of plant-based yogurt alternatives in Denmark and similar countries.

## 2. Theoretical Background

### 2.1. Theory of Planned Behaviour

The theory of planned behaviour (TPB) has been successfully applied in the past to understand and predict food-related behaviours, including healthy eating [22], dietary behaviour [23], innovative products such as functional food [24,25] and green food consumption [26]. Several previous studies have shown that a positive attitude, subjective norms and perceived behavioural control significantly influence the intention to consume plant-based dairy alternatives, and that intention is an immediate antecedent for consumption behaviour of plant-based products [27,28,29]. Based on the TPB assumptions, this study intended to examine Hypotheses 1–4: *Attitudes, subjective norms, perceived behavioural control (self-efficacy) and intention affect behaviour to consume plant-based yogurt alternatives.*

### 2.2. Extension of the TPB through Literature Review

The TPB has received much empirical support and is one of the most influential theories used to predict reasoned human behaviours. However, researchers have criticised the TPB for being limited when predicting more complex behaviours such as food choices [30,31]. Thus, researchers have extended the TPB with additional factors that affect intention, and including additional constructs in the TPB framework has shown improvement in the predictive power of the framework [32,33,34]. Further, adding relevant constructs or altering the causal path of the variables in the framework can deepen and broaden the TPB framework [32,35,36]. Hence, on the basis of the literature review, this study attempts to include three variables in the TPB framework in the case of plant-based yogurt alternatives, i.e., objective knowledge, perceived barriers and perceived sensory attributes.

Consumers’ knowledge has a significant influence on consumption of plant-based dairy alternatives [27] and consumption of plant-based diet [37]. Further, consumers must have a sufficient level of knowledge, based on reliable information, for information to have a positive impact on the consumer decision-making process regarding food choice [38]. Thus, apart from the TPB construct, this study examines Hypotheses 5–6: *Objective knowledge significantly affects attitudes and intention to consume plant-based yogurt alternatives.*

Previous studies have indicated taste preferences, unavailability of plant-based food, price and lack of cooking skills and information as a barrier to adoption of a plant-based diet [36,37,38,39,40]. Further, the sensory attributes including taste, aroma, mouth fullness and tastiness were key predictors for increased acceptability and intention to purchase plant-based products [39,40,41]. Based on these findings this study will examine Hypotheses 7–8: *Perceived barriers and perceived sensory attributes affect intention to consume plant-based yogurt alternatives.*

Socio-demographic characteristics, age [42], gender [39,43,44], education [39,45] and income [39] were associated with consumption of a plant-based diet. Hence, this study examines Hypothesis 9a–h: *Socio-demographic characteristics (H9a: age; H9b: gender; H9c: high level of education; H9d: full-time employment; H9e: high level of income; H9f: Greater Copenhagen) and lifestyle determinants (H9g: participants who do most of the shopping; H9h: participants with a vegan dietary pattern) have a significant association with intention to consume plant-based yogurt alternatives.*

Figure 1 depicts the proposed theoretical framework based on the literature review.

## 3. Methods and Materials

### 3.1. Questionnaire and Measurement Scale

A questionnaire-based online survey was conducted to assess consumers’ attitudes towards and objective knowledge regarding plant-based yogurt alternatives, as well as their experience and expectation when consuming plant-based yogurt alternatives. The questionnaire was developed in English and consists of 18 closed questions with 39 variables in total and is divided into two sections.

The first section consists of social-demographic characteristics (7 items: age, gender, living status, education, employment, income, region of residence) and lifestyle determinants (2 items: shopping and dietary pattern).

The second section consists of the proposed theoretical model constructs and items. These, along with their source of adoption, are presented in Appendix A. The proposed theoretical model items were measured on a 5-point Likert scale ranging from 1 (‘strongly disagree’) to 5 (‘strongly agree’), except for objective knowledge items that were measured by ‘True’, ‘False’ or ‘I don’t know’. The item measuring behaviour (consumption of plant-based yogurt alternatives) was based on self-reported consumption of plant-based yogurt alternatives with frequency of consumption measured by the following items: ‘How often do you consume plant-based yogurt alternatives?’ with measures of ‘Never’, ‘Less than once a month’, ‘Once a month’, ‘Once every 2 to 3 weeks’, ‘Once a week’, ‘Once every 2 to 3 days’, and ‘Every day’. Finally, the preferred source of plant-based yogurt was measured using ‘Soy’, ‘Coconut’, ‘Almond’, ‘Hemp’, ‘Cashew’, and ‘Other with an open-ended option’.

### 3.2. Participants and Design

According to previous studies, a sample size of 200 and above offers adequate statistical power for exploratory factor analysis and structural equation modelling analysis [46]. Further, a sample size-to-items ratio of 10 participants per construct item is necessary for model precision—‘the ability of the construct estimates to approximate true population values’ [47]. A convenience (snowball) sampling technique was used to recruit participants for the survey. The questionnaire was administered through the self-creation of a hyperlink generated by the Survey-Xact platform from 10 July 2020 to 15 August 2020. A brief description of the research was posted in different social media groups, namely ‘Plant-based food for beginners’, ‘Vegan Denmark’, ‘More focus to lactose intolerance’, ‘Lactose intolerant Denmark’, etc. Participants were asked for their consent regarding participation in the survey and were made aware of the time needed (approximately 5–8 min) prior to completing the questionnaire.

### 3.3. Data Management

The data sets stored by Survey-Xact were transferred in an anonymous manner to a file compatible with IBM SPSS Statistics 26 [48] for data management and statistical analysis. The fully completed questionnaire was included for further analysis. All reverse-scaled statements of the questionnaire were recorded in the same direction. Answers to the four objective knowledge statements regarding plant-based yogurt alternatives were re-coded as 1 for the correct answer and 0 for the wrong answer and the ‘I don’t know’ response. The final objective knowledge measure was computed as the total number of correct responses, thus ranging from 0 to 4. This computation was based on previous research [49].

### 3.4. Data Analysis

Descriptive statistics were based on shown percentages and numbers for categorical variables and median (interquartile range, IQR) for continuous variables. Exploratory factor analysis was performed for the items of proposed theoretical framework constructs through principal component analysis. Factor loadings of items less than standard (<0.70) were not assigned [50]. Varimax with Kaiser Normalization was applied to improve the interpretation of factors. The Kaiser, Meyer and Olkin (KMO) and Bartlett’s tests of sphericity were selected for the tests for goodness of fit of factor analysis. Cronbach’s alpha and composite reliability (CR) were used for evaluating constructs reliability [51] and Average Value Extract score (AVE) was selected for evaluating convergent validity [52]. Correlation coefficient between the constructs and the square root of the AVE were estimated to determine discriminant validity of the constructs. Further, the relationships between the proposed theoretical model constructs were assessed by using a structural equation model (SEM). The underlying assumptions of the model were based on the original TPB framework (direct path from attitude, subjective norms and perceived behavioural control to behavioural intention). Analysis of both the original TPB model and the proposed theoretical model (Figure 1) were conducted to determine whether the proposed theoretical model fits the data more accurately than the original TPB model. The model fit was reported by chi-square, the GFI (Goodness of Fit Index), TLI (Tucker Lewis Index), CFI (Comparative Fit Index) and RMSEA (Root Mean Square Error of Approximation). Further, the association between plant-based yogurt alternative products consumption behaviour and socio-demographic characteristics and lifestyle determinants was investigated by logistic regression.

In all statistical tests, *p*-values of less than 0.05 were considered statistically significant. SEM analyses were carried out using IBM SPSS AMOS 26 [53].

## 4. Result

### 4.1. Socio-Demographic Characteristics and Lifestyle Determinants

A sample size of 265 participants between the age range of 15–65 years old who consume plant-based yogurt alternatives completed the online survey. The results from Table 1 show that the sample was heterogeneous, however it was biased in terms of participants being predominantly female (80.8% of the participants), with full-time employment (41.5% of the participants), residing in the Greater Copenhagen region (70.6% of the participants), following a vegan diet (60.8% of the participants), and soy (57.0% of the participants) as a preferred plant-based yogurt alternative source. Regarding consumption frequency of plant-based yogurt alternatives, most of the participants consume plant-based yogurt alternatives every two to three days (21.2% of participants), while around 10.6% of the participants consume plant-based yogurt alternative on a daily basis.

### 4.2. Exploratory Factor Analysis and Reliability and Validity Tests

The results (Appendix A) show that the six factors (attitude, subjective norms, perceived behavioural control (self-efficacy), perceived barriers, perceived sensory attributes and intention) were extracted through Varimax with Kaiser Normalization rotation that explained 66% of the variance in the data. The Kaiser-Meyer-Olkin measure of sampling adequacy value equals 0.782, whereas Bartlett’s test of sphericity yielded a *p*-value of <0.001.

Cronbach’s alpha α ranged from 0.720 for perceived behavioural control (self-efficacy) to 0.939 for intention, indicating an acceptable homogeneity limit of >0.70 among the items of a respective construct [51]. Further, the AVE score ranged from 0.559 for perceived barriers to 0.766 for subjective norms, indicating above acceptable limit of >0.50 [54]. The CR value ranged from 0.735 for perceived behaviour control (self-efficacy) to 0.907 for subjective norms, also indicating above acceptable limit of >0.60 [55]. Furthermore, the results from Appendix A indicates that the square root of the AVE was higher than the correlation coefficient between each construct, indicating good adequacy for discriminant validity [52]. All in all, the proposed theoretical model has adequate convergent and discriminant validity and reliability measures.

### 4.3. The Goodness of Fit Result

Structural analysis using the goodness of fit indices was performed to test the model fit of the proposed theoretical framework (Figure 1). The result from Appendix A show that the proposed theoretical framework has a good fit (ꭓ^2^ = 339.659, ꭓ^2^/df = 1.963, *p* < 0.000, GFI = 0.898, TLI = 0.924, CFI = 0.937, and RMSEA = 0.060), thus within the acceptable limits recommended by [55]. Further, when compared to the original TPB framework (direct path from attitudes, subjective norms and perceived behavioural control to intention), the proposed theoretical framework has better model fit indices and improved descriptive power for predicting intention to consume plant-based yogurt alternatives (adjusted R^2^ = 0.292 versus 0.225 for original TBP framework). Besides, the fit indices of the proposed framework were relatively better (χ^2^/df = 1.963, RMSEA = 0.060) that the original TPB framework (χ^2^/df = 3.954, RMSEA = 0.106). The results supported the inclusion of objective knowledge, perceived barrier and perceived sensory attributes in the TPB in the case of plant-based yogurt alternatives.

### 4.4. Hypothesis Testing Result

The results from structural analysis testing (Figure 2) show that attitude (β = 0.216, *p* < 0.001), perceived behavioural control (self-efficacy) (β = 0.229, *p* < 0.001) and perceived sensory attributes (β = 0.324, *p* < 0.001) were significantly related to intention to consume plant-based dairy alternative products. Hence, hypotheses H1, H3 and H8 were supported, respectively. However, subjective norms (β = 0.106, *p* = 0.087), objective knowledge (β = −0.029, *p* = 0.603) and perceived barrier (β = −0.051, *p* = 0.414) were not significant predictors of intention to consume plant-based dairy alternative products, rejecting hypotheses H2, H6 and H7, respectively. Further, objective knowledge (β = 0.077, *p* = 0.242) was not a significant predictor of attitudes towards plant-based yogurt, rejecting hypothesis H5. Intention (β = 0.677, *p* < 0.001) was a positive and significant predictor of behaviour to consume plant-based dairy alternatives, thus H4 was supported. Appendix A shows the paths, their standardised estimate and t-value, and hypothesis status.

Finally, the results from logistic regression (Table 2) indicate that gender, high level of education, high level of income and region of residence have a significant association with behaviour towards plant-based yogurt alternatives, supporting hypotheses H9b, H9c, H9e and H9f, respectively. Female participants were more likely to frequently consume plant-based yogurt alternatives than male participants (odds ratio = 2.860). Participants with a high level of education, high level of income and who reside in Greater Copenhagen were more likely to frequently consume plant-based yogurt alternatives than their counterparts (odds ratios of 2.856, 3.546 and 2.370, respectively). However, age, employment status, shopping and dietary pattern have no significant association with behaviour towards plant-based yogurt alternatives, thus rejecting hypotheses H9a, H9d, H9g and H9h, respectively.

**Figure 2 foods-10-00148-f002:**
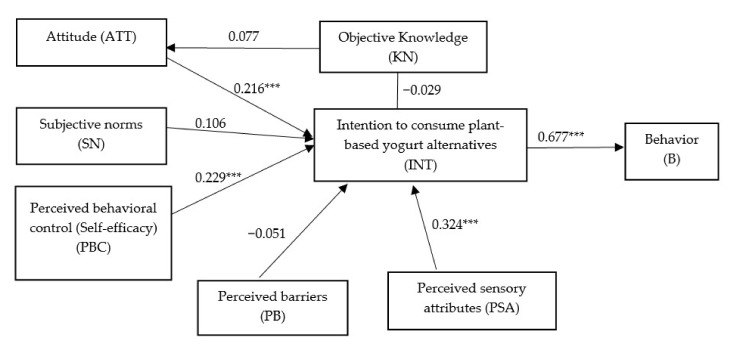
Structural relation between constructs (N = 265). Note: *** significant effect at *p* < 0.001.

## 5. Discussion and Implications

This exploratory study attempts to predict consumers’ intention to consume plant-based yogurt, primarily through Ajzen’s TPB framework along with three constructs: objective knowledge, perceived barriers and perceived sensory attributes. The results of this study strongly validate the sufficiency of the TPB framework in predicting plant-based yogurt consumption. The results unveil that attitude, perceived behavioural control (self-efficacy) and perceived sensory attributes considerably attribute to the development of intention to consume plant-based yogurt alternatives. However, subjective norms, perceived barriers and objective knowledge show no significant influence on the intention to consume plant-based yogurt alternatives. Among the TPB constructs, perceived behavioural control (self-efficacy) has emerged as the most significant factor in predicting consumers’ intention to consume plant-based yogurt alternatives (β = 0.229). This finding is consistent with a previous meta-analysis on health-related intentions and behaviour [56]. Further, this study was based on consumers who already consume plant-based yogurt alternatives, which might explain their high self-efficacy towards plant-based yogurt alternatives. The high self-efficacy among participants in this study might also be the explanation for perceived barriers not being a significant predictor of intention to consume plant-based yogurt alternatives. The finding is consistent with a study by [43]. The authors argue that plant-based alternative products are readily and easily available in Swedish supermarkets and that Swedish consumers “do not think about money first” and did not like the taste of dairy alternatives, which hinders them from purchasing it. This might be a possible explanation for the findings in this study, as the plant-based yogurt alternatives are easily available and accessible from supermarkets and online stores.

According to the findings from this study, attitudes towards plant-based yogurt alternatives are a positive and significant predictor of intention to consume plant-based yogurt alternatives. A similar finding was reported in previous studies [43,57]. Further, a recent study of European consumers found that the health and sustainability involved segment had strongly positive attitudes towards plant-based diets, with a mean score of 7.81 out of 10 [58]. The involvement of healthy and sustainable eating was also previously associated with a diet containing a higher proportion of plant-based foods [59]. This might be a possible explanation for positive attitudes towards plant-based yogurt alternatives among plant-based yogurt consumers in Denmark. Consumer’s strong attitudes towards plant-based yogurt alternatives indicate that producers and marketers need to understand that consumers are more concerned about the environment, sustainability, and their health. Thus, they should include such features that are apparent and appeal to the consumer.

According to the findings of this study, subjective norms did not have a positive and significant influence on the intention to purchase plant-based yogurt alternative products. This finding was inconsistent with several previous studies [27,43,45,60].

The findings from this study indicate that increasing objective knowledge regarding plant-based yogurt alternatives does not influence consumers’ attitude and intention towards plant-based yogurt alternatives. Worsley [61] suggests that knowledge alone is not sufficient to have an actual impact on eating habits; however, knowledge is positively linked to dietary intake [62]. A recent study of European consumers found that consumers had limited knowledge of plant-based diet composition; however, it found that consumers’ objective knowledge can be an important contributor to support perceptions of plant-based diets in comparison with vegetarian and vegan diets [37]. Further, Alba and Hutchinson [63] found that consumers are generally overconfident about themselves, thus their overall level of subjective knowledge tends to be greater than their objective knowledge, expressed for instance as the percentage of the correct response to a list of factual knowledge questions. Similarly, the author of [49] found that objective knowledge had a relatively weak effect on attitudes, and its effect on consumption behaviour is mediated by general attitudes and subjective knowledge.

The findings from this study indicate that perceived sensory attributes are strong and significant predictors for intention to consume plant-based yogurt alternatives (β = 0.324). Several consumer studies have indicated sensory attributes, especially taste, as an important determinant factor for plant-based diets [37,64,65]. However, sensory attributes can vary regarding the plant-based source, for instance, plant-based yogurt based on soy had a higher texture, coconut had the highest odour and preference was distributed between coconut, soy and cashew [2]. A study by Kumar and Thakur [66] showed that adding banana to soy yogurt significantly increases flavour, consistency and overall consumer acceptance. Thus, plant-based yogurt alternatives producers should focus on and improve the organoleptic characteristics of these products and providing the consumer with clear and visible information about the sensory quality of plant-based yogurt alternatives is recommended.

As expected, behavioural intention was a significant and the most direct predictor of plant-based yogurt alternative consumption behaviour as well as the immediate antecedent of such behaviour [67]. These findings are consistent with findings from previous research on meat and dairy alternatives [43].

### Strengths and Limitations

This research is the first of its kind to apply the extended TPB model to understand consumers’ intention to consume plant-based yogurt alternatives. This research was conducted on individuals who consume plant-based yogurt alternatives, thus drivers and barriers to consumption behaviour regarding plant-based yogurt alternatives as well as the association between behaviour regarding plant-based yogurt alternatives and socio-demographic characteristics and lifestyle determinants were explored.

Despite its strengths, the research has some limitations. First, the sample size (N = 265) for the SEM analysis was just above the minimum requirement to test multiple hypotheses in the proposed theoretical model of interacting variables [68]. The cross-sectional design does not allow the inference of causality, but only provides suggestive tendencies. Further, the convenience sampling method employed for recruiting participants may have affected the generalisability of the findings of this study due to selection bias. This study was conducted among individual who consumes plant-based yogurt alternatives; thus the majority of the respondents were following vegan dietary patterns (60.8% of the respondents). A structured questionnaire was utilised to collect data, which restrained important insights into consumers’ consumption decisions. Limited previous studies on plant-based yogurt alternatives have restricted the comparison and contrasting of results to research on a plant-based diet, plant-based dairy substitutes and plant-based foods. Consumer behaviour regarding plant-based yogurt alternatives was measured using self-reported measures, which may have resulted in over- or under-reporting of consumption frequency [69]. This study only considered the influence of objective knowledge and two additional factors: perceived barriers and perceived sensory attributes. Other important factors such as health consciousness [18,57], environmental concern [57,70] and subjective knowledge [49] might have strengthened the understanding of consumers’ decision-making process regarding plant-based yogurt alternatives as well as improved the predictive power of the proposed theoretical model.

## 6. Conclusions and Future Research

In conclusion, the study has empirically applied the TPB framework with three additional constructs: objective knowledge, perceived barriers and perceived sensory attributes. The additional constructs have increased the robustness and predictive ability of the proposed theoretical framework when predicting consumers’ intention to consume plant-based yogurt alternatives. Overall, attitude, perceived behavioural control (self-efficacy) and perceived sensory attributes were significant predictors for the intention to consume plant-based yogurt alternatives among plant-based yogurt consumers in Denmark. However, subjective norms and perceived barriers failed to show a significant influence on the intention to consume plant-based yogurt alternatives. Further, objective knowledge showed no significant influence on attitudes and intention towards plant-based yogurt consumption. Furthermore, female participants with high education and income levels, who reside in Greater Copenhagen, were more likely to consume plant-based yogurt alternatives.

Future studies with larger and statistically representative consumer samples and a longitudinal investigation in a cross-national context are recommended. Future studies can also apply the segmentation approach together with the TPB framework to better understand the complexities of food choice regarding yogurt consumption.

## Figures and Tables

**Figure 1 foods-10-00148-f001:**
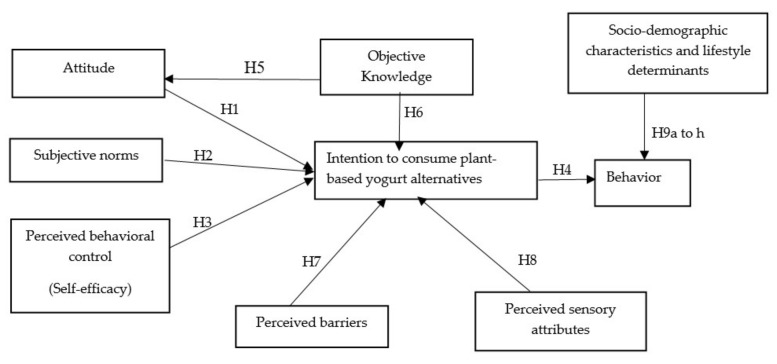
Proposed theoretical framework.

**Table 1 foods-10-00148-t001:** Socio-demographic characteristics and lifestyle determinants, N = 265 ^a^.

Variables	Categories	
Age, y, median (IQR)		29 (14)
Gender, %(n)	Male	19.2 (51)
	Female	80.8 (214)
Living, %(n)	Alone	16.2 (43)
	With partner	30.9 (82)
	Partner with children	24.9 (66)
	Single parent	5.3 (14)
	With parents	5.3 (14)
	With roommates	14.3 (38)
	Other	3.0 (8)
Education, %(n)	Primary school	6.0 (16)
	High school/vocational	15.8 (42)
	Bachelor	35.1 (93)
	Masters and above	40.0 (106)
	Other	3.0 (8)
Employment, %(n)	Student	13.6 (36)
	Student with a job	14.0 (37)
	Part-time	9.8 (26)
	Full-time	41.5 (110)
	Self-employed	7.5 (20)
	Unemployed	12.8 (34)
	Retried	0.8 (2)
Income, %(n)	DKK 10,000 or under	10.2 (27)
	DKK 10,001 to DKK 15,000	6.8 (18)
	DKK 15,001 to DKK 25,000	11.3 (30)
	DKK 25,001 to DKK 35,000	18.9 (50)
	DKK 35,001 to DKK 45,000	9.8 (26)
	DKK 45,001 and above	27.9 (74)
	No statement	15.1 (40)
Region of residence, %(n)	Greater Copenhagen	70.6 (187)
	Outside Greater Copenhagen	29.4 (78)
Shopping, %(n)	I do most of the shopping	61.5 (163)
	I do 50% of the shopping	27.2 (72)
	I sometimes do the shopping	9.1 (24)
	I never do the shopping	2.3 (6)
Dietary pattern ^b^, %(n)	Omnivorous	14.3 (38)
	Vegetarian	12.8 (34)
	Flexitarian	12.1 (32)
	Vegan	60.8 (161)
Consumption frequency of plant-based yogurt alternatives, %(n)	Less than once a month	17.4 (46)
	Once a month	17.4 (46)
	Once every 2 to 3 weeks	17.0 (45)
	Once in a week	16.6 (44)
	Once every 2 to 3 days	21.1 (56)
	Everyday	10.6 (28)
Preferred plant-based yogurt alternative source, %(n)	Soy	57.0 (151)
	Coconut	15.1 (40)
	Almond	8.3 (22)
	Oats	7.5 (20)
	Others including cashew, hemp, peanuts, etc.	9.8 (26)
	No preference	2.3 (6)

**Note**: ^a^ Data is presented as median (interquartile range, IQR) or percentage (frequency); ^b^ dietary pattern: Omnivorous: eats everything, Vegetarian: avoids meat (and fish) but eats eggs and dairy products, Flexitarian: mainly vegetarian (occasionally meat consumption), Vegan: avoids all animal-based products.

**Table 2 foods-10-00148-t002:** Association between plant-based yogurt alternatives behaviour and socio-demographic characteristics and lifestyle determinants (N = 265).

Variables	Hypothesis	Odds Ratio (OR)	95% Confidence Interval	*p*-Value
Age (years)	H8a	0.990	0.963–1.019	0.503
Gender (Female = 1)	H8b	2.860	1.290–6.342	0.010 **
High-level education (1)	H8c	2.856	1.387–5.882	0.004 **
Full-time employment (1)	H8d	0.584	0.302–1.128	0.110
High-level income (1)	H8e	3.546	1.770–7.106	≤0.001 ***
Region (Greater Copenhagen = 1)	H8f	2.370	1.215–4.624	0.011 *
Shopping (I do most of the shopping = 1)	H8g	1.875	0.985–3.570	0.056
Dietary pattern (Vegan = 1)	H8h	1.730	0.898–3.334	0.101
Constant		0.002	–	≤0.001 ***

**Note**: * significant effect at *p* < 0.05, ** significant effect at *p* < 0.01, *** significant effect at *p* < 0.001. Logistic regression with the dependent variable as behaviour towards plant-based yogurt alternatives with consumption frequency ‘Once a week’, ‘Once every 2 to 3 days’ and ‘Everyday’ coded as 1 and consumption frequency ‘Less than once a month’, ‘Once a month’ and ‘Once every 2 to 3 weeks’ coded as 0. High-level education = Bachelor and above, High-level income = DKK 25,001 and above.

## Data Availability

The data presented in this study are available on request from the corresponding author. The data are not publicly available due to ethical reason.

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
