# Peer review of "An Application of the Theory of Planned Behaviour to Predict Intention to Consume Plant-Based Yogurt Alternatives"

_foods, 2021, doi:10.3390/foods10010148_

Round 1

Reviewer 1 Report

Foods-1057148 - Review: “An application of the theory of planned behaviour to predict Danish consumers’ intention to consume plant-based yogurt alternatives”

This is an interesting paper on examining Danish plant-based yogurt consumer’s intention to consume plant-based yogurt. The authors use of the Theory of Planned Behaviour for this analysis. They explain how the limitations of the theory can be tackled by incorporating new constructs into it. Although there are new expansions to the Theory of Planned Behaviour TPB (Reasoned Action Approach) that the authors could have used instead the used of the TPB is considered valid.

However, the title and large part of the manuscript seems to suggest that the analysis refers to all Danish consumers when this is not the case. The analysis refers to plant-based yogurt consumers. This makes the results very limiting and less interesting. In order to solve this issue the questionnaire should have been distributed to the wider population.

Comments:

  • The authors state that “subjective norms (β = 0.106, p = 0.087) … did not influence consumers’ intention to consume plant based yogurt alternatives”. Given the p-value being less than 0.10 this would be a statistically significant results at 10% significance level.
  • Page 1: The authors state that “As a result, the majority of consumers prefer a plant-based diet [3].” The reference refers to the US market and although the article highlights the fact that dairy milk consumption is declining, it is not clear that the majority of US consumers consume non-dairy milk to dairy milk. Hence, the statement seems too strong given the evidence provided.
  • Page 1: The authors state that “A study in Denmark by [6] showed that greenhouse gas emission was 48% higher for the average diet compared to a plant-based diet” This is correct. However, the cited paper also highlights that “when optimizing a diet with regard to sustainability, it is crucial to account for the nutritional value and not solely focus on impact per kg product. Excluding dairy products from the diet does not necessarily mitigate climate change but in contrast may have nutritional consequences.” In their discussion the authors clearly state that “This study shows that excluding dairy products from our diet does not necessarily mitigate climate change; however, it may have nutritional consequences”. I would suggest the authors to balance their message.
  • Page 1: The authors continue by stating: “The shift towards a plant-based diet significantly improves food sustainability and environmental impact [7,8] and also has beneficial health implications.” Here the authors say that the shift towards a plant-based diet has beneficial health implications. This statement as it stands is too strong given it is opposite to what Werner et al. (2014) say in their research.
  • Could the authors be more specific on how product developers and/or others can benefit from these results? Is there any suggestions that can help?
  • Section 3.2 Participants and design: The online survey seemed to have been distributed through channels that reach a population of consumers that already may be inclined to consume plant-based yogurt alternatives. The results of this analysis will therefore refer to this population and not the typical Danish consumer. This information needs to be clearly stated before. The title of the article may lead to confusion since it refers to Danish consumers, which is true in the sense that the consumers are Danish but are not representative of typical Danish consumer. This point is briefly discussed on the discussion “Further, this study was based on consumers who already consume plant-based yogurt alternatives, which might explain their high self-efficacy towards plant-based yogurt alternatives.” However results from this study cannot be extrapolated to the general Danish consumer. One option to overcome the problem could be to distinguish (if possible) between plant based consumers and non-plant based consumers, integrate this information in the model and rerun the analysis.
  • Page 8: The authors find that “increasing objective knowledge regarding plant-based yogurt alternatives does not influence consumers’ attitude and intention towards plant-based yogurt alternatives”. This finding is somehow not surprising given the sample used.
  •  

Author Response

Thank you so much for your comments.

Point 1:The authors state that “subjective norms (β = 0.106, p = 0.087) … did not influence consumers’ intention to consume plant based yogurt alternatives”. Given the p-value being less than 0.10 this would be a statistically significant results at 10% significance level.

Response 1: In section 3.4 data analysis, page 5, we have mentioned that p-values of less than 0.05 were considered statistically significant in all statistical tests.

Point 2: The authors state that “As a result, the majority of consumers prefer a plant-based diet [3].” The reference refers to the US market and although the article highlights the fact that dairy milk consumption is declining, it is not clear that the majority of US consumers consume non-dairy milk to dairy milk. Hence, the statement seems too strong given the evidence provided.

Response 2: We have formatted the sentence as:- As a result, there is a growing shift towards a plant-based diet

Point 3: The authors state that “A study in Denmark by [6] showed that greenhouse gas emission was 48% higher for the average diet compared to a plant-based diet” This is correct. However, the cited paper also highlights that “when optimizing a diet with regard to sustainability, it is crucial to account for the nutritional value and not solely focus on impact per kg product. Excluding dairy products from the diet does not necessarily mitigate climate change but in contrast may have nutritional consequences.” In their discussion the authors clearly state that “This study shows that excluding dairy products from our diet does not necessarily mitigate climate change; however, it may have nutritional consequences”. I would suggest the authors to balance their message.

Response 3: The health effects of dairy according to the recent paper on the New England Journal of Medicine's recent paper by World’s leading Nutrition Epidemiologist Prof. Walter Willett and Dr. David Ludwig concluded that dairy might be “nice to have” but not necessary from nutrition and health perspectives. Even if it can be acceptable in places of chronic malnutrition and hunger, the threats of its consumption outweigh the benefits. Therefore, we have presented the result of the study by Werner et al. (2014) rather than focusing on nutritional aspect between them. Moreover, the aforementioned article has been funded by Arla AMB, one of the largest dairy companies in Europe, which in itself is a conflict of interest (not disclosed by the authors), and it needs to be taken cautiously when authors specifically address the role of dairy in sustainability, and brings the nutritional value into discussion.

Point 4: The authors continue by stating: “The shift towards a plant-based diet significantly improves food sustainability and environmental impact [7,8] and also has beneficial health implications.” Here the authors say that the shift towards a plant-based diet has beneficial health implications. This statement as it stands is too strong given it is opposite to what Werner et al. (2014) say in their research.

Response 4: There are plenty of article supporting that plant-based diet has a beneficial health implication. We have presented meta-analysis by Fan and colleagues (2019). Other references are as follows:

Kim at al. (2019) https://doi.org/10.1161/JAHA.119.012865

Satija et al. (2016)   https://doi.org/10.1371/journal.pmed.1002039

Point 5: Could the authors be more specific on how product developers and/or others can benefit from these results? Is there any suggestions that can help?

Response 5: The knowledge from this study contributes to and extends further understanding of consumers’ consumption behaviour of plant-based yogurt alternatives, identifying the rationales for consuming plant-based yogurt alternatives. The findings on consumers’ consumption behaviour can be valuable for food industry when generating new products by directly integrating consumers’ perception of plant-based yogurt alternatives. Furthermore, the identification of consumers key drivers and barriers to consume plant-based yogurt alternatives can contribute to assess the market context.

Point 6: Section 3.2 Participants and design: The online survey seemed to have been distributed through channels that reach a population of consumers that already may be inclined to consume plant-based yogurt alternatives. The results of this analysis will therefore refer to this population and not the typical Danish consumer. This information needs to be clearly stated before. The title of the article may lead to confusion since it refers to Danish consumers, which is true in the sense that the consumers are Danish but are not representative of typical Danish consumer. This point is briefly discussed on the discussion “Further, this study was based on consumers who already consume plant-based yogurt alternatives, which might explain their high self-efficacy towards plant-based yogurt alternatives.” However results from this study cannot be extrapolated to the general Danish consumer. One option to overcome the problem could be to distinguish (if possible) between plant based consumers and non-plant based consumers, integrate this information in the model and rerun the analysis.

Response 6: Our study aims to understand factors influencing behavioural intention and behaviour to consume plant-based yogurt alternative products, thus the sample is justified. And, one of the limitation of our study is that the findings cannot be generalised to Danish consumer, which is already stated on the limitation of the study.

The aim of the study is revised as "to predict factors influencing intention to consume plant-based yogurt alternative products among plant-based yogurt consumers in Denmark".

Point 7: Page 8: The authors find that “increasing objective knowledge regarding plant-based yogurt alternatives does not influence consumers’ attitude and intention towards plant-based yogurt alternatives”. This finding is somehow not surprising given the sample used.

Response 7: That might be the case and including subjective knowledge might be relevant to consider in future studies. And this is also included in limitation of this study.  

Reviewer 2 Report

The paper is well presented and the results are properly interpreted, including limitations of the study. However, some minor concerns may arise: why was the questionnaire in English and how did the Authors check that people filling it have a good understanding of the termini included (e.g. "mouthfeel")? Another, rather unusual feature is the sampling method, i.e. people were recruited and asked through the internet. Please, give some references supporting the validity of such online questionnaires used for similar purposes.

Author Response

Thank you so much for your comments. Below are some explanation to the comments on “Why was the questionnaire in English and how did the Authors check that people filling it have a good understanding of the termini included (e.g. "mouthfeel")? Another, rather unusual feature is the sampling method, i.e. people were recruited and asked through the internet. Please, give some references supporting the validity of such online questionnaires used for similar purposes.”

  • The participants were recruited through snowball sampling. There are several other previous researches that has used similar approach (Faber et al., 2020; Reipurt et al., 2019). Regarding the validity, we are aware of the potential bias in this kind of methodology, therefore it has been mentioned in the limitation.

Reviewer 3 Report

However, no research has been conducted to study the factors influencing consumer intention to consume plant-based yogurt alternatives using the theory of planned behaviour (TPB).

The paper has , in my opinion, an adequate level of originality.

It can be considered as an exploratory study and aims to investigate both  the key drivers and both the barriers to consuming plant-based yogurt alternatives.

The authors applied the TPB framework by adding additional constructs

In paragraph 5.1. (Strengths and limitations) the limitations are clearly indicated,

I required a few formal corrections and suggested the addition of citiations in the attached file (see notes I added).

I read the document with great interest.

Author Response

Thank you so much for your comments.

I have revised the comments you have made; however, the following statements are from my findings from the analysis and therefore there is no reference.

“This might be a possible explanation for positive attitudes towards plant-based yogurt alternatives among Danish consumers. Danish consumer’s strong attitudes towards plant-based yogurt alternatives indicate that producers and marketers need to understand that consumers are more concerned about the environment, sustainability, and their health. Thus, they should include such features that are apparent and appeal to the Danish consumer.”

Round 2

Reviewer 1 Report

Thank you for your response. 

On points 3 and 4 my suggestion was to balance the message, which authors do not seem to fully agree with. I understand that there is literature that suggests that plant-based diet may have beneficial health implications. However, this is not what the authors are saying. I believe the authors could have made their case without using strong statements such as "The shift towards a plant-based diet significantly improves food sustainability and environmental impact [8,9] and also has beneficial health implications [10–12]." The definition food sustainability concept is a debatable issue. Werner et al. (2014) state that "combining nutritional value and sustainability aspects ... is one step toward finding a more accurate way to address sustainable food consumption." This would indicate that food sustainability is multidimensional including not only food production, but also environmental and health with potential trade-offs between them. For instance, findings by Vieux et al. (2020) suggest that low dietary GHGE may lead to low nutritional quality. Another important aspect that has not been mentioned is that although there is an increase in plant based diet consumption it is not clear whether this will be a sustained increase. The papers cited (e.g. [8]) also find that majority of consumers "in developed markets like Austria, there is still a clear, positive product image of cow milk as a natural and healthy product.". 

Finally, on point 6. My understanding is that the sample consists of consumers that are already plant based yogurt consumers. This means that what is analysed is the plant based yogurt consumer's intention and behaviour to consume plant based yogurt alternatives. Still, the message in the paper is confusing "Overall, attitude, perceived behavioural control (self-efficacy) and perceived sensory attributes were significant predictors for the intention to consume plant-based yogurt alternatives among Danish consumers."

References:

Vieux et al. (2020) More sustainable European diets based on self-selection do not require exclusion of entire categories of food, Journal of Cleaner Production

Author Response

Thank you for your comments.

Response to point 3-4: 

The text has been revised as "The shift towards a plant-based diet may significantly improves food sustainability and environmental impact [8,9] and also has beneficial health implications [10–12]." 

Response to point 6: 

The text has been revised as "Overall, attitude, perceived behavioural control (self-efficacy) and perceived sensory attributes were significant predictors for the intention to consume plant-based yogurt alternatives among plant-based yogurt consumers in Denmark."
